# Knowledge Atlas on the Relationship between Water Management and Constructed Wetlands—A Bibliometric Analysis Based on CiteSpace

Yanqin Zhang , Xianli You, Shanjun Huang, Minhua Wang and Jianwen Dong *

College of Landscape Architecture, Fujian Agriculture and Forestry University, Fuzhou 350002, China;
zyq97120530@163.com (Y.Z.); yxianli0709@163.com (X.Y.); hsj7174@163.com (S.H.); wmh699@163.com (M.W.)
* Correspondence: fjdjw@fafu.edu.cn; Tel.: +86-136-0952-5156

**Abstract:** Water management is a crucial resource conservation challenge that mankind faces, and encouraging the creation of manmade wetlands with the goal of achieving long-term water management is the key to long-term urban development. To summarise and analyse the status of the research on the relationship between water management and constructed wetlands, this paper makes use of the advantages of the bibliometric visualization of CiteSpace to generate country/region maps and author-collaboration maps, and to analyse research hotspots and research dynamics by using keywords and literature co-citations based on 1248 pieces of related literature in the core collection in the Web of Science (WoS) database. The existing research shows that the research content and methods in the field of constructed-wetland and water-management research are constantly being enriched and deepened, including the research methods frequently used in constructed wetlands in water management and in the research content under concern, the functions and roles of constructed wetlands, the relevant measurement indicators of the purification impact of constructed wetlands on water bodies, and the types of water bodies treated by constructed wetlands in water management. We summarise the impact pathways of constructed wetlands on water management, as well as the impact factors of constructed wetlands under water-management objectives, by analysing the future concerns in the research field to provide references for research.

**Keywords:** water management; sustainable development; knowledge map; quantitative analysis

## 1. Introduction

Water is a significant component of the UN Sustainable Development Goals (SDGs). Over 80% of wastewater is discharged without treatment directly into the environment around the world [1]. A total of 2.2 billion people lack access to safe drinking water [2]. Extreme climate and rapid urbanization have resulted in worldwide water depletion and an increased demand for water resources, and have altered the natural water cycle, despite the fact that the water cycle is intimately linked to the healthy development of cities [3]. The main reason for the water shortage is that the demand exceeds the supply. With population growth, urbanization, and socioeconomic development, the demand for water for urban industry and domestic use is expected to increase by 50–80% over the next 30 years [4]. In fact, water is an indispensable resource for the future development of urban communities, and it is necessary to find ways to implement the sustainable development of water resources in urban spaces [5]. Cities must integrate the urban design process with other disciplines that are responsible for water management in order to achieve sustainable urban water management (SAWM) [6]. Nature-based solutions (NBSs) are an effective and cost-efficient approach to sustainable urban water development [7]. NBSs are green infrastructures [8], and, over the long term, NBS green infrastructures have been frequently more effective at increasing urban resilience than grey infrastructures [9]. As a result,

nature-based solutions are becoming more prevalent in policy creation and urban planning guidelines [10].

Combining constructed wetlands (CWs) with low-cost green infrastructure provides an innovative method for urban water management [11]. Constructed wetlands offer the ability to address the current urban-water-management issues by treating grey water, sewage-treatment-plant effluent, and industrial wastewater [1,12–14]. The treatment approach to CWs is low-maintenance and low-operational, and it enhances the natural landscape while providing various benefits [15]. Compared with 37.1% for a cyclic activated sludge system, and 28.1% for a hypothetical case of conventional activated sludge, the constructed wetland has a minimal power consumption of 3.9%, which significantly reduces maintenance resources [16]. The treatment steps in wastewater-treatment plants are divided into nitrification and denitrification, followed by the mechanical dehydration of the productive sludge to 80%, and then transport to a sanitary landfill. However, CWs treat wastewater by controlling the timing, duration, and direction of the water flow. In addition, the $CO_2$ emissions from wastewater-treatment plants are almost seven times higher than those from vertical-subsurface-flow CWs [17]. Moreover, CWs are a significant technological approach to sustainable water management because of their limited maintenance [18,19]. Simultaneously, urban planners and landscape architects are better able to use CWs and integrate them into the built environment. Nevertheless, at this stage, the public is less concerned about the impact of the function of CWs on urban water management and is more inclined to focus on their biological value, comfort, aesthetics, etc. [14]. As a result, we focus on the function of CWs in water management, as well as on future research into novel approaches for updating, using, managing, and sustaining traditional CWs. This study is based on the methodology, substance, and results of the existing related literature, and we integrated it with CiteSpace software to conduct a literature visualization econometric analysis of the literature to evaluate the influence of CWs on water management, and to examine the status of the CWs, research hotspots, and research developments in the context of sustainable urban water management. The key research questions are as follows:

1.  What are the publication statuses and growth trends in the field of water-management and CW research?
2.  Which countries/regions and authors have influenced water-management and CW research?
3.  In the field of water-management and CW research, what are the research keywords and essential literature?

In this study, we assessed the literature and its posts in the field of water-management and CW research published between 2002 and 2022, and we then performed a more precise and particular analysis based on the country/region of the research area and author-collaboration-network mapping. This study also includes a complete analysis and description of the keywords (distribution, timeline, and occurrence) to better reflect the research hotspots and directions, as well as the interactions between the research hotspots and time ranges involved. On this foundation, we evaluate the CW impact pathways and mechanisms on water management, as well as describe the CW impact factors in relation to water-management goals. Finally, this paper proposes a research direction to achieve the sustainable development of water management through CWs.

## 2. Materials and Methods

### 2.1. Data Sources

Retrieved by (TS = (constructed wetlands) AND TS = (water management)), the data source for the study was the Web of Science (WoS) Core Collection, the time span was 2002–2022, and the retrieval time was 5 May 2022. The document type was journal articles in the English language. This study employed a total of 1248 publications as its data source.

## 2.2. Research Method

Scientific-knowledge mapping is a new research tool in the subject of scientometry and informatics that uses a graphical language to describe a field's research process, knowledge structure, and evolution trend. CiteSpace takes as input a set of bibliographic records and models of the knowledge structure of the underlying domain based on a synthetic network of network timeseries derived from annual publications. CiteSpace supports many types of bibliometric studies, including collaborative-network analysis, co-word analysis, author-co-citation analysis, document-co-citation analysis, and textual and geospatial visualization [20]. CiteSpace was created by Prof. Chaomei Chen, and its bibliographic and visualization functions can present the development trends and knowledge association statuses of disciplinary frontiers in a very intuitive way, which can quickly grasp the key information of the research field [20–22]. To comprehend the status of the research and research hotspots in the research field, researchers undertake quantitative calculations based on the strengths of the correlations between terms and cluster them. Keywords are the lifeblood of a paper, describing the primary information while also offering the shortest possible summary of the content [23]. We use keyword analysis in order to gain insight into the main features of a field, and to enable a reasonable description of the research frontiers and future research directions [24].

We used CiteSpace (5.8.R3) (Figure 1) to quantitatively assess the relevant literature, and to construct a corresponding knowledge map based on the literature in the field of water-management and CW research. In addition, we identified the current state and hot spots of the research, and we summarise the functions of CWs from the perspective of water management. Moreover, we elaborate on water-management and CW impact mechanisms, as well as on the research field's development trend, with the goal of providing a reference and basis for future research.

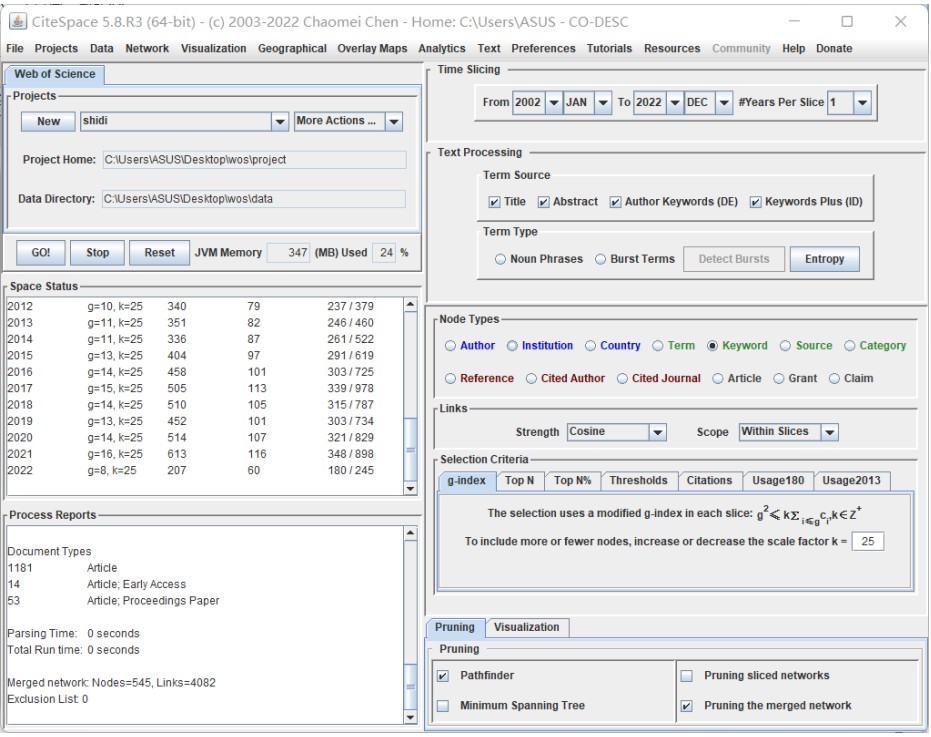

**Figure 1.** The main user interface of CiteSpace.

## 3. Basic Situation Analysis

### 3.1. Trends in the Number of Published Papers

We analysed a total of 1248 publications that were distributed between 2002 and 2022. Research on CWs from the perspective of water-resource management has grown steadily

over the last two decades. Figure 2 shows that, between 2002 and 2022, the number of studies in the subject area in the Web of Science database increased year by year (note: the main types of literature are articles), and the research history can be divided into three stages: (1) the initial stage, from 2002 to 2008, when CW research moved closer to water-resource management, with an annual number of articles between 20 and 40; (2) the accumulation phase, from 2009 to 2014, when the annual number of publications remained relatively stable, ranging between 40 and 70 articles in 2009, with 70 articles in 2013; (3) the steady-growth phase, from 2015 to 2020, when the volume of literature steadily increased each year, with the number fluctuating between 80 and 100 articles, with 102 articles in 2017. The number of publications increased to 140 in 2021.

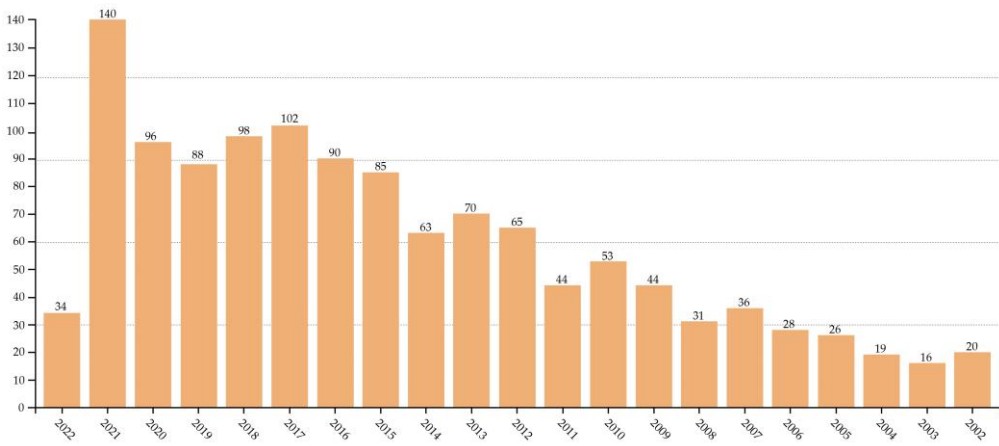

**Figure 2.** Schematic diagram of document quantities.

### 3.2. Cooperation Networks

### 3.2.1. Country/Region Cooperation Networks

An analysis of the collaborative networks between countries/regions allows for the identification of priority countries/regions that have generated large numbers of publications and that have had a significant impact on the research field, as well as for the identification of the collaborative relationships between them. We found that, from 2002 to 2022, a total of 89 countries were involved in the field of water-management research on constructed wetlands, with an intertwined network of partnerships. Figure 3 shows a collaboration-network map of the countries/regions created with CiteSpace software. Pathfinder and pruning the merged network are the pruning in Figure 3. The following information can be derived from the analysis of the data: the total number of network nodes is 89 (N = 89), the nodes are connected by 133 links (E = 133), and the density of the research-field network is 0.034 (density = 0.034). The circles in the diagram represent frequencies, and the sizes of the circles are proportional to the frequency counts. The lines that connect different nodes denote the presence of many nodes in the same literature at the same time. The centrality indicates the significance of a particular node in the network [25]. Thus, the greater the centrality, the greater the influence of the posting in those countries/regions.

Figure 3 and Table 1 demonstrate that the United States leads in the number of studies (counts = 407). scientific institutions make the most outstanding contributions to the research. To begin, researchers in the United States concentrated on removing pollutants such as nitrogen and phosphorus from CWs [26]. Then, they investigated the elements that influence the CWs' purification rates, and modelled the purification rates [27,28]. Moreover, research was conducted on the role of CWs in urban water management [29].

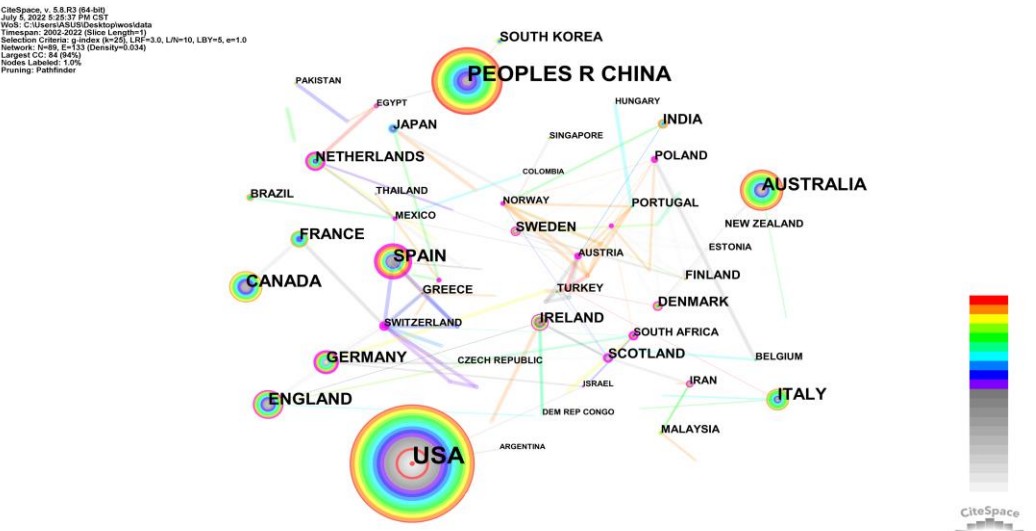

**Figure 3.** Collaboration-network map of countries/regions.

**Table 1.** The top 10 countries/regions in frequency and centrality between 2002 and 2022.

| Rank | Countries/Regions | Counts | Centrality | Year | Countries/Regions | Counts | Centrality | Year |
|------|-------------------|--------|------------|------|-------------------|--------|------------|------|
| 1 | United States | 407 | 0.09 | 2002 | Switzerland | 11 | 0.66 | 2003 |
| 2 | Peoples R China | 220 | 0.04 | 2006 | Austria | 10 | 0.47 | 2010 |
| 3 | Australia | 86 | 0.04 | 2003 | Scotland | 25 | 0.38 | 2003 |
| 4 | Spain | 66 | 0.26 | 2004 | Germany | 42 | 0.32 | 2002 |
| 5 | Canada | 65 | 0 | 2002 | Mexico | 10 | 0.29 | 2004 |
| 6 | England | 64 | 0.13 | 2002 | Netherlands | 32 | 0.28 | 2005 |
| 7 | Italy | 55 | 0.09 | 2002 | Poland | 20 | 0.27 | 2002 |
| 8 | France | 42 | 0.09 | 2006 | Spain | 66 | 0.26 | 2004 |
| 9 | Germany | 42 | 0.32 | 2002 | South Africa | 13 | 0.26 | 2003 |
| 10 | Netherlands | 32 | 0.28 | 2005 | Norway | 10 | 0.23 | 2005 |

The United States is followed by China in second place in terms of the number of articles published (counts = 220). There are macroscopic studies on the CW dispersal in China [30–32], as well as research into the efficiency of the integrated management of CW media, plant management, etc. [33–35]. Although the number of studies is large, the impact is relatively lacking (centrality = 0.04).

The Switzerland literature ranked first in terms of impact (centrality = 0.66), which indicates that its findings are globally worthy of being referenced by scholars. The literature proposed CWs combined with phytoremediation techniques [36,37], as well as a macrostudy of CWs in water management [10,38,39].

### 3.2.2. Author-Cooperation Networks

Between 2002 and 2022, 586 authors contributed to this field of study. Figure 4 shows a collaboration-network map of the authors created with CiteSpace software. The software defaults to each setting. The following information could be acquired from the analysis of the data: the total number of network nodes is 587 (N = 587), the nodes are connected by 319 links (E = 319), and the density of the research-field network is 0.0019 (density = 0.0019). Many isolated points could be illustrated in the figure, and only a few authors have collaborative networks with each other. Table 2 shows the top 10 authors in terms of the number of articles published. M. Scholz (counts = 10) is the most productive author in terms of the number of publications, followed by C M Cooper (counts = 8), R. Kroeger (counts = 6), R. Harrington (counts = 5), and S. A White (counts = 5). Miklas Scholz started maintaining a certain number of publications in 2006, including a study

on the efficiency of CW water at removing pollutants from agricultural wastewater over a seven-year period [40], as well as the rates of agricultural and domestic wastewater purification by an integrated constructed wetland simulated by the multiple regression model, principal component analysis, redundancy analysis, and a self-organizing map model. Moreover, the study investigates the recycling of grey water treated by CWs for crop irrigation, and the high temperature to improve the quality of the water purification in CWs [41,42]. Robert Kroeger's research focuses on a pollutant breakdown by plant [43–45], Rory Harrington's research focuses on integrated CW research [40,46], and Sarah A White's research focuses on floating-wetland purification [47].

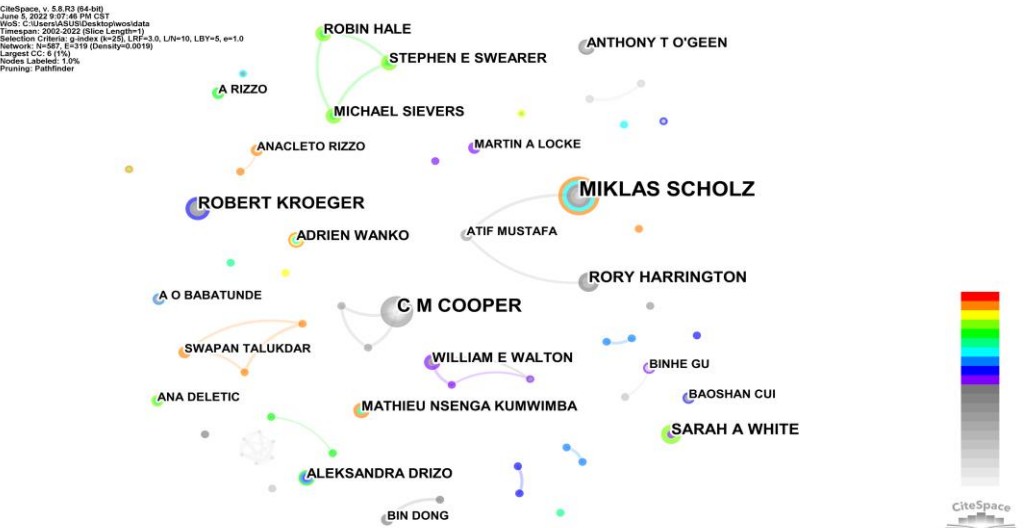

**Figure 4.** Collaboration-network map of authors.

**Table 2.** The top 10 authors and co-cited references between 2002 and 2022.

| Rank | Authors | Counts | Year |
|------|---------|--------|------|
| 1 | Miklas Scholz | 10 | 2006 |
| 2 | C M Cooper | 8 | 2002 |
| 3 | Robert Kroeger | 6 | 2011 |
| 4 | Rory Harrington | 5 | 2009 |
| 5 | Sarah A White | 5 | 2013 |
| 6 | Robin Hale | 4 | 2018 |
| 7 | Aleksandra Drizo | 4 | 2012 |
| 8 | Mathieu Nsenga Kumwimba | 4 | 2017 |
| 9 | Adrien Wanko | 4 | 2009 |
| 10 | Anthony T O'Geen | 4 | 2009 |

## 4. Knowledge-Base Analysis

### *4.1. Keyword Analysis*

#### 4.1.1. Keyword Co-Occurrences

The pruning parameter (pruning: pathfinder and pruning the merged network) was used to create the keyword-co-occurrence-network map: 545 nodes and 1036 links were included in the network, and the density was 0.007 (Figure 5). There was no analysis for "constructed wetland "and" management" because the search query contained "constructed wetland "and" management." Table 3 illustrates the top 20 (of 543) keywords in terms of frequency and centrality between 2002 and 2022. The keywords removal (214) and performance (137) had the highest frequency counts, followed by wastewater (134), water (132), and nitrogen (123). Moreover, the keywords with the highest centrality were wastewater treatment (0.17), water quality (0.16), and community (0.16). According to

Table 3, the keywords related to improving the water quality (82/0.16) include wastewater treatment (103/0.17) and fresh water (24/0.1). Attention will also be paid to the effects of its different plant communities (27/0.16) on the water-purification rates, and the performance (137/0.01) of the wetland use, even after including the value of CW ecological services, such as conservation (36/0.11) and service (8/0.11).

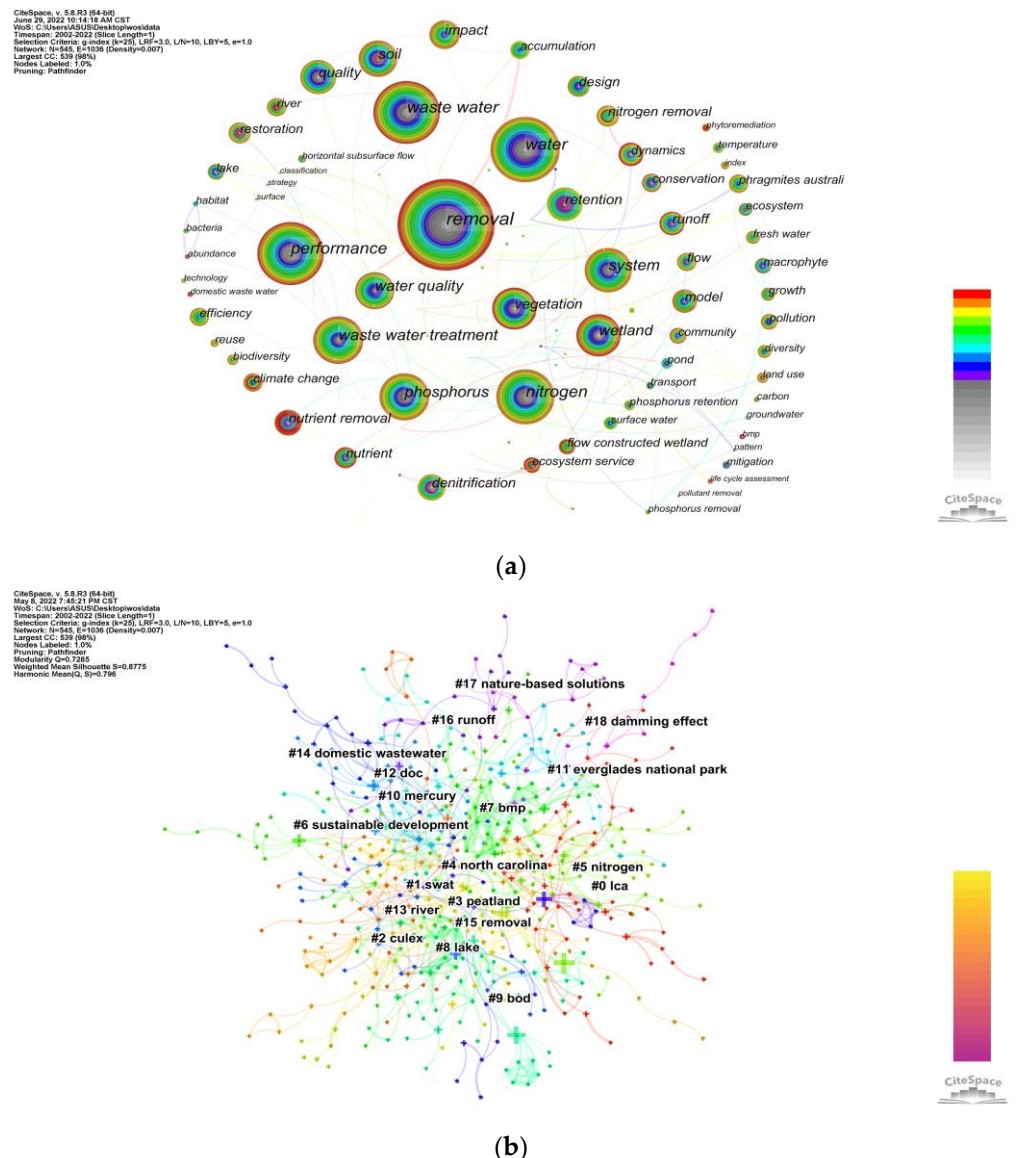

(**a**)

(**b**)

**Figure 5.** (**a**) Keyword-co-occurrence-visualisation map of publications on "constructed wetlands "water management" from 2002 to 2022. The circles represent the frequency, and the circle sizes are positively correlated with the frequency counts. The lines between different nodes indicate that different nodes appear in the same literature simultaneously. (**b**) Cluster map of keywords.

**Table 3.** Top 20 keywords in frequency and centrality between 2002 and 2022.

| Rank | Keywords | Counts | Centrality | Year | Keywords | Counts | Centrality | Year |
|------|----------|--------|------------|------|----------|--------|------------|------|
| 1 | removal | 214 | 0.04 | 2004 | wastewater treatment | 103 | 0.17 | 2004 |
| 2 | performance | 137 | 0.01 | 2006 | water quality | 82 | 0.16 | 2003 |
| 3 | wastewater | 134 | 0.06 | 2006 | community | 27 | 0.16 | 2005 |
| 4 | water | 132 | 0.06 | 2002 | denitrification | 53 | 0.13 | 2003 |
| 5 | nitrogen | 123 | 0.1 | 2003 | conservation | 36 | 0.11 | 2005 |

**Table 3.** *Cont.*

| Rank | Keywords | Counts | Centrality | Year | Keywords | Counts | Centrality | Year |
|------|----------|--------|------------|------|----------|--------|------------|------|
| 6 | wastewater treatment | 103 | 0.17 | 2004 | atrazine | 12 | 0.11 | 2004 |
| 7 | system | 96 | 0.03 | 2006 | adsorption | 12 | 0.11 | 2008 |
| 8 | phosphorus | 94 | 0.01 | 2004 | ammonia | 10 | 0.11 | 2003 |
| 9 | water quality | 82 | 0.16 | 2003 | decomposition | 9 | 0.11 | 2004 |
| 10 | wetland | 79 | 0.05 | 2003 | service | 8 | 0.11 | 2013 |
| 11 | soil | 68 | 0.05 | 2003 | artificial wetland | 4 | 0.11 | 2002 |
| 12 | quality | 67 | 0.07 | 2003 | nitrogen | 123 | 0.1 | 2003 |
| 13 | vegetation | 67 | 0.04 | 2005 | catchment | 25 | 0.1 | 2003 |
| 14 | retention | 66 | 0.03 | 2005 | fresh water | 24 | 0.1 | 2002 |
| 15 | impact | 61 | 0.05 | 2004 | bmp | 11 | 0.1 | 2003 |
| 16 | denitrification | 53 | 0.13 | 2003 | nutrient | 47 | 0.09 | 2009 |
| 17 | runoff | 52 | 0.04 | 2005 | accumulation | 37 | 0.09 | 2003 |
| 18 | nutrient removal | 52 | 0.07 | 2005 | carbon | 17 | 0.09 | 2004 |
| 19 | sediment | 52 | 0.01 | 2005 | area | 10 | 0.09 | 2002 |
| 20 | model | 49 | 0.03 | 2006 | metal | 11 | 0.08 | 2009 |

The log-likelihood rate (LLR) was utilised to cluster the keyword network in this paper, and the nouns of the feature words with the highest LLR operator values are employed as the cluster names. A modularity (Q) and mean silhouette of 0.7285 and 0.8775, respectively, are shown in Figure 5b, which demonstrate a considerable clustering structure and somewhat accurate and persuasive clustering results. The 19 main clusters are shown in Figure 5b, and the information about the 19 keyword-co-occurrence clusters is shown in Table 4. The 19 clusters were divided into four categories to better comprehend the research material of the subject area, and to better understand the cluster-analysis results:

- The first category focuses on the research methods and research content often utilised in water-management research in wetlands and constructed wetlands. These include: #0 (LCA), #1 (SWAT), #2 (culex), #3 (peatland), #4 (North Carolina), and #11 (Everglades National Park). Valerie J. Fuchs acquired CWs through LCA with less environmental impact with regard to resource consumption and greenhouse gas emissions [48]. Several of the best management practices (BMPs), such as filter strips, grassed rivers, constructed wetlands, and detention basins, were also evaluated using the soil and water assessment tool (SWAT) model [49,50]. Culex and peatland are the hotspots of the study. North Carolina and Everglades National Park were early adopters of the use of constructed wetlands in water management, and many studies have focused on them [51,52];

- The second category, which includes Clusters #6 (sustainable development) and #17 (nature-based solutions), represents the function and role of manmade wetlands. Constructed wetlands (CWs) are an environmentally friendly and reliable green technology for treating all types of water bodies. They retain a high potential for application and are a key element for sustainable water management [53]. Moreover, the potential of constructed wetlands (CWs) to provide a wide range of ecosystem services as green infrastructure, with even higher benefits than grey infrastructure in water-management applications, is key to implementing natural solutions in water management [54,55];

- The third category focuses on the indicators of the CW water-purification indicators, such as #5 (nitrogen), #9 (BOD), #10 (mercury), #12 (DOC), and #15 (removal). The primary focus of this research is on the nitrogen-removal effectiveness of various types of built wetlands [56]. The BOD and DOC are often utilised to measure the efficiency of the constructed-wetland action. Indicators such as mercury [57], heavy metals [58], and microplastics [59] are also gradually incorporated into the study of the water purification in constructed wetlands;

- The fourth category is the collection of water types cleansed by CWs. There is growing interest in constructed wetlands, which are used to treat all sorts of wastewater (#14 (domestic wastewater)), are the best management practices for stormwater (#7 (BMP),

#16 (runoff), and #18 (damming effect)), and are even used to treat urban ecological waters (#8 (lake), #13 (river)).

**Table 4.** Information on the 19 largest co-occurrence clusters of keywords.

| Cluster | Size | Silhouette | Mean (Year) | Label (LLR) |
|---|---|---|---|---|
| 0 | 48 | 0.923 | 2014 | Life cycle assessment (LCA) |
| 1 | 42 | 0.816 | 2013 | Soil and water assessment tool (SWAT) |
| 2 | 39 | 0.843 | 2010 | Culex |
| 3 | 38 | 0.819 | 2012 | Peatland |
| 4 | 35 | 0.87 | 2009 | North Carolina |
| 5 | 35 | 0.925 | 2008 | Nitrogen |
| 6 | 33 | 0.892 | 2012 | Sustainable development |
| 7 | 32 | 0.929 | 2006 | Best management practices (BMPs) |
| 8 | 31 | 0.919 | 2009 | Lake |
| 9 | 29 | 0.856 | 2007 | Biochemical oxygen demand (BOD) |
| 10 | 28 | 0.832 | 2008 | Mercury |
| 11 | 27 | 0.839 | 2013 | Everglades National Park |
| 12 | 25 | 0.919 | 2010 | Dissolved organic carbon (DOC) |
| 13 | 22 | 0.878 | 2013 | River |
| 14 | 20 | 0.763 | 2015 | Domestic wastewater |
| 15 | 17 | 0.93 | 2011 | Removal |
| 16 | 16 | 0.917 | 2012 | Runoff |
| 17 | 12 | 0.978 | 2016 | Nature-based solutions |
| 18 | 10 | 0.939 | 2014 | Damming effect |

4.1.2. Keyword-Trend Analysis

CiteSpace generated keyword-time-zone maps that focus on representing the evolution of the research hotspots in the field from a temporal dimension. A timeline analysis of keywords focuses on revealing the relationship between the clustering and the historical span of keyword sets. Keyword outbursts can reflect the changes in research topics and hotspots in a field. Therefore, in this paper, we started from these three perspectives to conduct a keyword-trend analysis in this research area.

The evolutionary trend of keywords in this research area from 2002 to 2022 is roughly divided into three phases. The pruning parameters (prune: pathfinder, pruning sliced networks, and a prune merged network) and the top N = 30 were selected to create Figure 6. The network contained 111 nodes and 169 links, with a density of 0.0277 (Figure 6). By combining the nodal years of the keyword appearances, the 20-year development of this research field was analysed, as shown in Figure 6. The phase when the high-frequency keywords in this research field increased dramatically was 2002–2006, and the papers mainly focused on water-quality improvement and wastewater purification. In this phase, the exploration of the use of CWs for rainwater, industrial-wastewater, and agricultural-wastewater purification began. As early as 2022, a hog farm in North Carolina investigated the effectiveness of artificial wetlands in treating swine wastewater from an anaerobic lagoon [60]. Sylvia Toet et al. demonstrated that CWs can effectively improve the water quality of wastewater-treatment-plant effluent [61]. In addition, A.M. Ibekwe et al. tentatively uncovered that 50% plant-covered CWs can purify water more effectively [62]. In the next phase (2007–2016), CWs were excavated more deeply to find the effects of different variables of CWs on water purification, such as plants, temperature, water flow, and dynamics. For example, AK Upadhyay et al. found that the application of a simulated CW was more effective at solving water-pollution problems, and that submerged plants accumulated large amounts of toxic elements in the CW [63]. Moreover, focusing on the uptake of heavy metals in CWs, Anna Guittonny-Philippe et al. showed that the order of the plant uptake of heavy-metal elements was Fe > Al > Mn > Cr, Ni, Zn > Cu > As, Cd, Pb [64]. In addition, the functions of additional ecosystem services of CWs were found. In the final phase (2017–2022), the land-use area of CWs was considered comprehensively,

and the focus was on evaluating the life cycle of CWs. Yongqiu Xia and Xiaoyuan Yan show that optimizing the location of CWs is a top priority for water-quality management and yields greater benefits than expanding the area of CWs elsewhere [65]. In the future, the use of CWs as a green infrastructure will face a number of challenges. Tiantao Zhou et al. show that CWs can help to mitigate urban flooding, but there are important considerations for implementation, such as the scale and capacity involved, the adequacy of the urban space, and their economic sustainability [66].

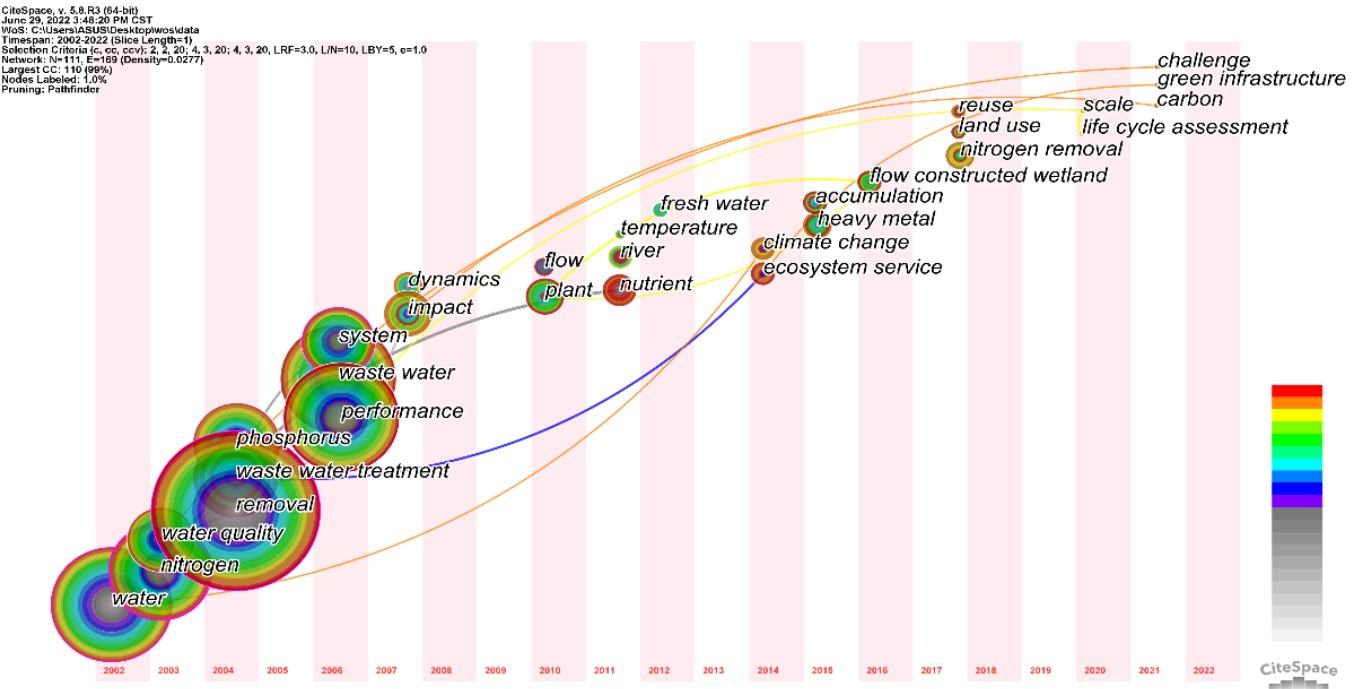

**Figure 6.** Research keyword time-zone-view map.

In order to be able to visually present the evolution and development trends of keyword clusters and keyword word sets in different time periods, a timeline visualisation of keywords was conducted. By analysing and summarising the keywords of each cluster, 19 hot topics were found. Figure 7 shows the timeline diagram of the research in the field of CWs and water management, with the horizontal coordinates of time, showing the evolution of keywords in the time dimension, which could clearly and intuitively discover the research trends. Figure 7 explains the characteristics of the research in this research area, as well as the mainstream research in each phase from 2002 to 2022.

Two clusters that became significant at the beginning of the study, and that have remained important for a long time, are #5 (2002–2022) and #12 (2002–2022). Cluster #5 is a key indicator of how effective CWs are at their jobs. The importance of CWs in eliminating nitrogen and phosphorous, the selection of plants, and CW monitoring that uses multiple models are all research themes in this cluster. For example, the potential of two free-surface CW systems to reduce pesticide concentrations in surface water was evaluated by combining in-field monitoring and dynamic fugacity modelling [67]. Recently, research on CWs has focused on complex agroecosystems [68], and nitrogen and phosphorus removal is no longer a frontier or hot spot. The #12 research was still ongoing in the early stages of the project (2002–2004) to investigate the impact of different designs on the doc factors, such as plants, media, bedding, etc. After it was found that phytoremediation had a greater impact on the doc factors, it appeared that combining CWs with phytoremediation techniques, such as selecting Potamogeton crispus and Hydrilla verticillata as plants for CWs to remove pollutants from water, was found to be more effective at solving water-pollution problems in the middle of the study [63].

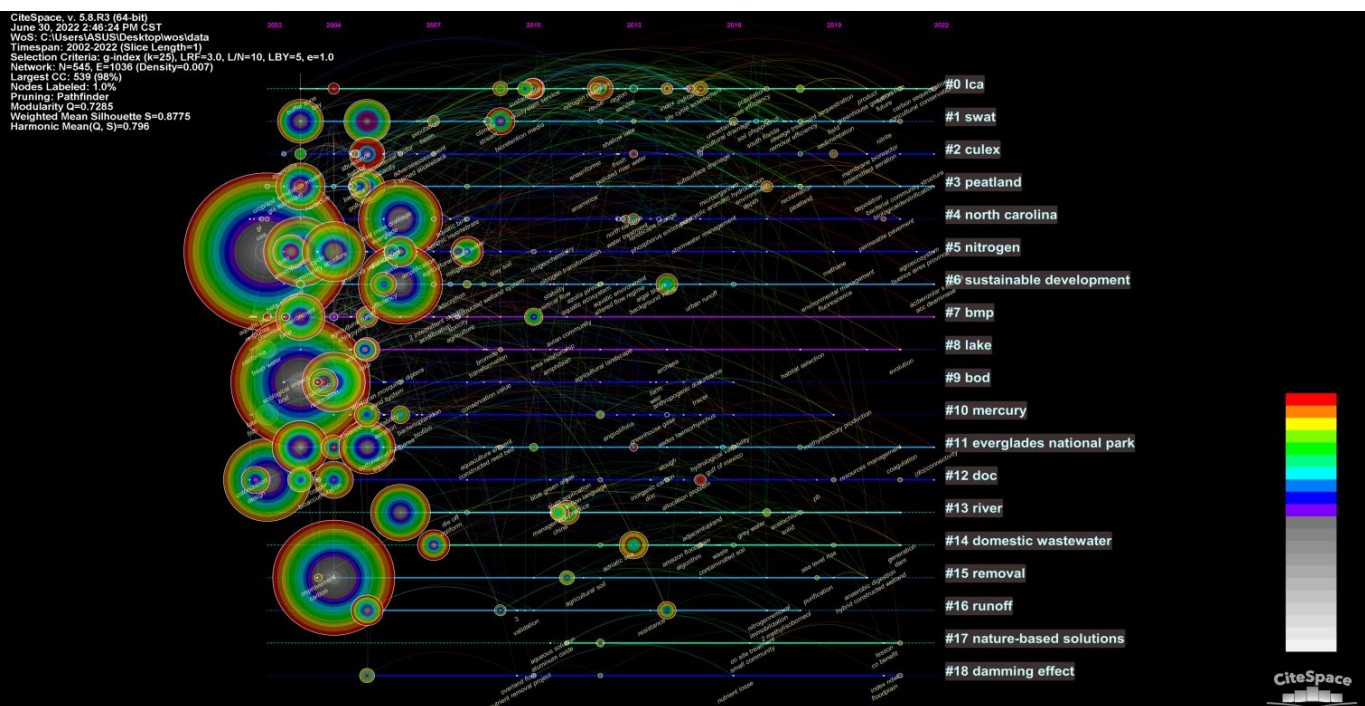

**Figure 7.** Research-keyword-timeline map.

Moreover, #0 and #1 are the important assessment methods for wetlands and constructed wetlands. The study of the #0 tendency to assess the function and characteristics of CWs and wetlands, as well as the clustering, was more active in the middle term (2010–2016), when the focus was on "flow constructed wetland", "ecosystem service", "sustainability", "biodiversity", "climate change", and "land use" research. Cluster #1 is focused on evaluating the water-purification capacity of wetlands and CWs, and its research keywords are most concentrated in the early stage (2009–2009) and mainly include "quality", "retention", and "nutrient".

Cluster #6 (sustainable development (2005–2019)) reflects the function and role of constructed wetlands. The study of #6 began in 2002, and it has remained a significant cluster for a long time. CW technology has been utilised to treat a wide range of water conditions as a developing cost-effective and sustainable method [69,70]. The research themes in this cluster were mainly focused on "waste water", "efficiency", "reduction", and "metal" in the early stages (2003–2009), and expanded to "heavy metal", "stability", and "aquatic environment" in the midterm (2010–2016). Recently, the research has gradually focused on "environment management" and "acibenzolar s methyl", which is the combination of low-environmental-risk fungicides, phytohormones, and other compounds in CWs in order to improve the performance of CW vegetation in removing pollutants [71].

Finally, Cluster #14 (2007–2021) emerged late, and researchers gradually realised the usefulness of CWs for wastewater treatment [72]. Although the cluster is also interested in CW dynamics and nitrogen-removal effectiveness, at the later stage of the investigation, the cluster innovated and adapted CWs to the type of wastewater and the location of application (for instance, the innovative wall-cascade constructed-wetland (WCCW) system for the treatment of grey water in urban environments, and the automated CW-microbial fuel cell (MFC) power-management system (PMS) [73,74]).

Burst keywords are terms that have quick increases in frequency in CiteSpace, and keyword-burst analysis is a useful tool for identifying study directions that have received a lot of attention over time [75]. Figure 8 shows the top 25 keywords with the highest citation bursts. Analysed from the keyword bursts, early on, CWs were a potential best management practice (2003–2013), and properly designed CWs acquire phosphorus retention (2005–2011)

and thus mitigate water pollution from atrazine use (2004–2014). In the medium term, CWs were used for wastewater treatment, the absorption of heavy metals in water, dealing with soils (2007–2009), nonpoint source pollution (2010–2016), and rivers (2014–2015), with an emphasis on the restoration (2011–2013) role of implementing CWs, with a focus on phytoremediation (2015–2016), as well as an emphasis on the ecological benefits, such as the landscape (2015–2017), brought about by CWs. Recently, research on the use of CWs to combat eutrophication (2019–2022) has resurfaced, with a focus on the purification efficiency, comparing the efficiency of the nitrogen removal (2019–2022) and phosphorus removal in CWs with different structures, applicability ranges/scales, life cycles, etc., and on finding an inextricable relationship between CWs and land use (2019–2022) and climate change (2020–2022), as well as more studies on CWs from a macroscopic perspective, such as at the urban and watershed scales. In terms of the keyword strength, ecosystem services (8.11) and nitrogen removal (5.42) have nonnegligible positions in the study of CWs from the perspective of water management, which demonstrates that the study of the ecosystem-service functions of CWs, as well as their nitrogen-removal role, hold a place in the future.

## Top 25 Keywords with the Strongest Citation Bursts

| Keywords | Year | Strength | Begin | End | 2002 - 2022 |
|---|---|---|---|---|---|
| usa | 2002 | 3.58 | **2002** | 2010 | |
| denitrification | 2002 | 4.34 | **2003** | 2011 | |
| bmp | 2002 | 4.27 | **2003** | 2013 | |
| atrazine | 2002 | 4.08 | **2004** | 2014 | |
| culicidae | 2002 | 3.14 | **2004** | 2012 | |
| ecosystem | 2002 | 4.5 | **2005** | 2006 | |
| phosphorus retention | 2002 | 3.85 | **2005** | 2011 | |
| soil | 2002 | 4.03 | **2007** | 2009 | |
| flow | 2002 | 3.85 | **2009** | 2012 | |
| nonpoint source pollution | 2002 | 3.35 | **2010** | 2016 | |
| mitigation | 2002 | 4.39 | **2011** | 2014 | |
| restoration | 2002 | 3.12 | **2011** | 2013 | |
| retention | 2002 | 4.31 | **2012** | 2014 | |
| river | 2002 | 4.19 | **2014** | 2015 | |
| phytoremediation | 2002 | 4.45 | **2015** | 2016 | |
| landscape | 2002 | 3.42 | **2015** | 2017 | |
| flow constructed wetland | 2002 | 3.11 | **2016** | 2020 | |
| nitrogen removal | 2002 | 5.42 | **2019** | 2022 | |
| nitrate removal | 2002 | 4.13 | **2019** | 2022 | |
| eutrophication | 2002 | 4.12 | **2019** | 2020 | |
| land use | 2002 | 3.91 | **2019** | 2022 | |
| nutrient removal | 2002 | 3.15 | **2019** | 2020 | |
| ecosystem service | 2002 | 8.11 | **2020** | 2022 | |
| climate change | 2002 | 4.66 | **2020** | 2022 | |
| life cycle assessment | 2002 | 3.14 | **2020** | 2022 | |

**Figure 8.** Top 25 keywords with the strongest citation bursts.

### 4.2. Co-Cited-Reference Analysis

#### 4.2.1. Co-Cited-Reference Analysis

An important indicator is co-citation-literature analysis. Figure 9 is the Co-citation-cluster map of cited references. The pruning parameter (pruning: none) was used to create the co-cited-reference-cluster map, which had 921 nodes and 2851 links in the network, and a density of 0.0067 (Figure 9). There are 23 clusters of co-cited references, which are labelled by the LLR: #0 (excess nitrogen), #1 (wastewater treatment), #2 (water-quality consequence), #3 (floating treatment wetland), #4 (long-term performance), #5 (nature-based solution), #6 (stormwater detention area), and so on. We focus our analysis on the first seven clusters:

- The first largest cluster (#0 (excess nitrogen)) has 71 individuals and a silhouette value of 0.868, with 2010 as the average year. The other labels include wetland mesocosm, pesticide mixture, artificial runoff event, and mitigating agrichemical. The most relevant citer to the cluster is Lizotte, Richard E, Jr. (2012), who examined the mitigation efficiency of managed wetlands using agrochemicals [76];

- The second largest cluster (#1 (wastewater treatment)) has 71 members and a silhouette value of 0.9, with 2015 as the average year. Surface flow, integrated ecological treatment system, plant-harvest management, and rural wastewater are all part of the cluster. The most relevant citer to the cluster is Marzo, A (2018), who found that a hybrid wetland system can treat civil wastewater [77];
- The third largest cluster (#2 (water-quality consequence)) retains 62 members and has a silhouette value of 0.904, with 2015 as the average year. The label also has southeastern coastal plain, large agricultural watershed, restoring wetland hydrology, and surface-water nitrogen. The most relevant citer to the cluster is Bernhardt, Emily S (2008), who suggested paying attention to and effectively managing surface-water nitrogen loads [78];
- The fourth largest cluster (#3 (floating treatment wetland)) has 56 members and a silhouette value of 0.926, with 2017 as the average year. The cluster also contains environmental protection and assessments of the nitrogen, plant species, and phosphorus-removal potential. The most relevant citer to the cluster is Martinez-Guerra, Edith (2020), who reviews the role of wetlands in wastewater treatment, stormwater management, and pollutant removal [79];
- The fifth largest cluster (#4 (long-term performance)) has 54 members and a silhouette value of 0.947, with 2007 as the average year. The cluster also contains the treatment of farmyard runoff, livestock-wastewater management, statistical modelling, and contaminant removal. The most relevant citer to the cluster is Mustafa, Atif (2009), who studied the performance of the integrated-constructed-wetland (ICW) system in improving the water quality in the Annestown Creek Watershed, Ireland, from 2001 to 2007 [40];
- The sixth largest cluster (#5 (nature-based solution)) has 52 members and a silhouette value of 0.982, with 2018 as the average year. Studies on the energy–food nexus, urban case studies, theoretical concepts, and post-COVID-19 agri-food supply chains are also included in the cluster. The most relevant citer to the cluster is Carvalho, Pedro (2022), who suggested that built wetlands are part of a nature-based solution to the water–energy–food nexus [7];
- The seventh largest cluster (#6 (stormwater detention area)) has 47 members and a silhouette value of 0.93, with 2013 as the average year. The cluster also contains a t-shifting nutrient sink, source function, event-scale nutrient attenuation, and a hybrid surface–subsurface flow system. The most relevant citer to the cluster is Adyel, Tanveer M (2017.0), who points out that mixed CWs are more capable of cleaning stormwater pollutants than single-stage CWs [80].

Table 5 lists the top ten (out of 919 total) co-cited references from 2002 to 2022, ranked by count. The most highly "co-cited reference" was a book written by Kadlec R H and Wallace S. in 2008 (counts = 21, cluster = 4) [81], which was cited 8749 times, based on statistics from Google Scholar. In this research area, this book represents a wetland treatment study that was systematised to reveal the relationship between constructed wetlands and water management. This was followed by a review of constructed wetlands for wastewater treatment by Haiming Wu et al. (counts = 17, cluster = 1) [82]. Vymazal J studied the rates of nitrogen and phosphorus removal by CWs and reviewed them separately (counts = 17, cluster = 4) [83], which was followed by a summary of the types of CWs and their uses in different periods (counts = 15, cluster = 0) [84], as well as the use of CWs to remove pesticides from agricultural runoff species, which was summarised in 2015 (counts = 12, cluster = 1) [85]. The standard method for the examination of water and wastewater published by the American Public Health Association (APHA), the American Water Works Association (AWWA), and the Water Environment Federation (WEF) has the highest centrality (centrality = 0.33, cluster = 4) (WEF) [86].

**Table 5.** Top 10 co-cited references in frequency between 2002 and 2022.

| Rank | Cited References | Counts | Centrality | Year | Cluster |
|------|------------------|--------|------------|------|---------|
| 1 | Kadlec RH, 2009, TREATMENT WETLANDS, V0, P0 | 21 | 0.03 | 2009 | 4 |
| 2 | Baird RB, 2017, STANDARD METHODS EXA, V0, P0 | 18 | 0.13 | 2017 | 2 |
| 3 | Wu HM, 2015, BIORESOURCE TECHNOL, V175, P594, DOI 10.1016/j.biortech.2014.10.068 | 17 | 0.17 | 2015 | 1 |
| 4 | Vymazal J, 2007, SCI TOTAL ENVIRON, V380, P48, DOI 10.1016/j.scitotenv.2006.09.014 | 17 | 0.09 | 2007 | 4 |
| 5 | APHA/AWWA/WEF, 2017, STANDARD METHODS EXA, V23th, P0 | 16 | 0.33 | 2017 | 0 |
| 6 | Vymazal J, 2011, ENVIRON SCI TECHNOL, V45, P61, DOI 10.1021/es101403q | 15 | 0.2 | 2011 | 0 |
| 7 | Pavlineri N, 2017, CHEM ENG J, V308, P1120, DOI 10.1016/j.cej.2016.09.140 | 14 | 0.05 | 2017 | 3 |
| 8 | Scholz M, 2007, WETLANDS, V27, P337, DOI 10.1672/0277-5212(2007)27[337:TICWIC]2.0.CO;2 | 14 | 0.21 | 2007 | 4 |
| 9 | Vymazal J, 2015, ENVIRON INT, V75, P11, DOI 10.1016/j.envint.2014.10.026 | 12 | 0.02 | 2015 | 1 |
| 10 | Diaz FJ, 2012, AGR WATER MANAGE, V104, P171, DOI 10.1016/j.agwat.2011.12.012 | 12 | 0.15 | 2012 | 0 |

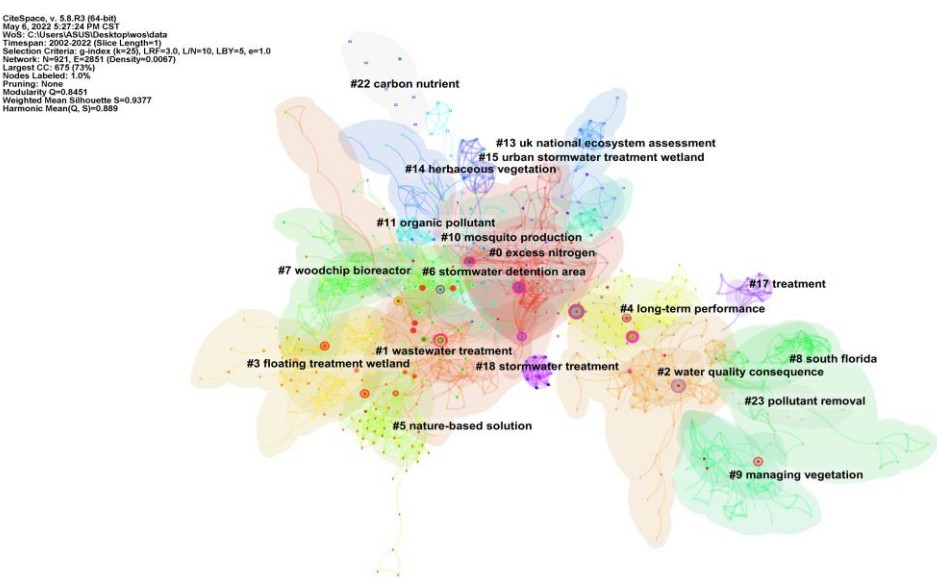

**Figure 9.** Co-citation-cluster map of cited references.

### 4.2.2. Co-Cited-Reference-Trend Analysis

Figure 10 depicts how the network is separated into distinct co-citation clusters over time. Clusters #3 and #5 are recently new research directions, with a citation explosion in the period 2016–2020. Cluster #3 contains observations on the removal performances of large mixed-incident urban wetlands (HAUWs) on urban streams, and a study of the nitrogen- and phosphorus-removal potentials of five plant species [87,88]. It demonstrates that floating treatment wetlands are a novel type of CW that are frequently deployed. In Cluster #5, Vasileios Takavakoglou et al. show opportunities for the application of CWs in different segments of the agri-food supply chain [89], and Joana AC Castellar et al. propose the use of nature-based solutions combined with advanced technologies for the decentralised recycling of urban greywater, of which CWs are an integral part. This shows

that nature-based solutions are gaining acceptance, and that CWs are a critical step in implementing natural water-management solutions.

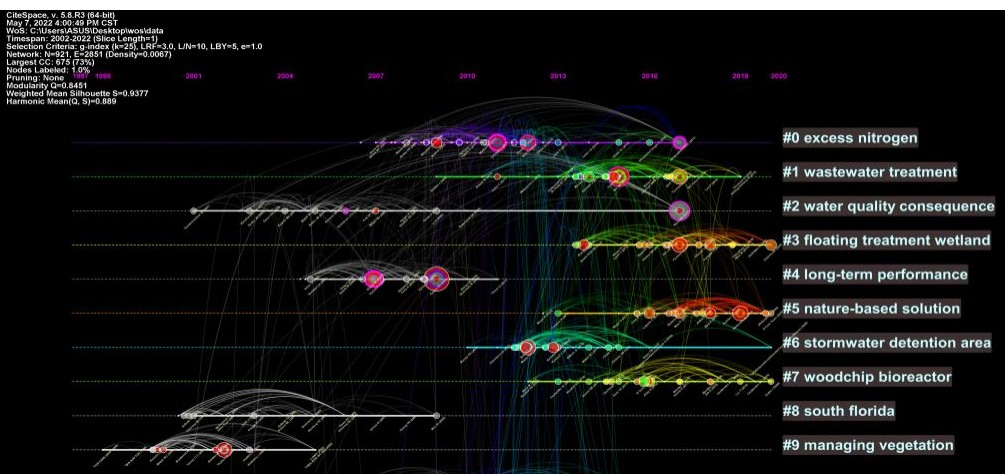

**Figure 10.** Co-citation-timeline map of cited references.

The publications in Clusters #0 and #2 have a stable number of citations over time, and with citation detonation. Clusters #1, #6, and #7 are more evenly cited and have been in the field of study for approximately a decade. The research areas of Clusters #4, #8, and #9 are of short durations and have low numbers of citation-triggered publications.

## 5. Impact Pathways of CWs on Water Management

There are three main impact pathways of CWs on water management, and Table 6 demonstrates studies related to the impact pathways of CWs on water management. CWs can reduce the urban flood risk and improve the urban environmental health by increasing stormwater storage [90], showing stormwater runoff, improving stormwater quality, and regulating municipal wastewater. Plant roots and substrates in CWs reduce the water flow and remove pollutants through biochemical reactions. Thivanka Dharmasena modelled the performances of CWs and measured that they effectively retain 62% of the stormwater throughout the year [91]. Stormwater runoff, agricultural runoff, and other pollutants migrate into cities and contain a variety of pollutants, including nutrients (e.g., nitrogen, phosphorus), potentially toxic metals (e.g., Pb, Zn, Cu, Cd), organic chemicals, pathogens, and suspended particles [92], which makes stormwater management critical to urban development. Table 6 demonstrates that CWs could effectively decrease the total suspended solids (TSS), total nitrogen (TN), total phosphorus (TP), chemical oxygen demand (COD), nitrate nitrogen ($NO_3{}^--N$), ammonium nitrogen ($NH_3-N$), ammonia nitrogen ($NH_4{}^+-N$), heavy metals, faecal coliform bacteria, perfluorooctanoic acid (PFOA), and perfluorooctane sulfonic acid (PFOS) in stormwater and municipal wastewater (Table 6), in order to play a positive role in the subsequent urban water management and simultaneously improve the health of the urban water environment.

The main causes of the physical, chemical (nutrients, metals, organics, nanomaterials, etc.), and microbial pollution of river water include untreated solid waste, stormwater, municipal sewage, agriculture runoff, and industrial wastewater that enter the river [93]. CWs were studied accordingly for landscape water bodies, sewage-treatment-plant discharge water, rivers, and streams (Table 6). The indicators that are often studied are the dissolved organic carbon (DOC), dissolved organic nitrogen (DON), and five-day biochemical oxygen demand ($BOD_5$). There are also combinations of several types of CWs for use, which include new tidal wetlands and submerged CWs [94], or tandem ecological floating beds, horizontal submerged CWs, and surface-flow constructed-wetland-treated urban rivers, with average removal rates of 74.79%, 80.90%, 71.12%, 78.44%, and 91.90% for COD, NH4+-N, TN, TP, and suspended solids (SS), respectively [95]. In terms of the

pollutant-removal efficiency, these CW technologies are significantly less expensive to employ than standard wastewater-treatment methods, and they are a green and sustainable technology that facilitates ecological water purification.

Constructed wetlands increase access to municipal wastewater treatment and increase the water-management method and efficiency. CWs are a good wastewater-treatment method that are based on natural solutions and have been applied to treat grey water, agricultural wastewater, industrial wastewater, etc. Not only do they save a lot of energy, but they can also purify tainted water to meet nonpotable reuse regulations [96]. First, grey water is domestic wastewater from sinks, laundry rooms, and showers that is heavy in food residues and that has significant volumes of grease and high concentrations of chemicals, sodium, phosphorus, surfactants, nitrogen, and non-biodegradable fibres from clothing [97,98]. At this stage of the study, wastewater treatment was conducted by floating treatment wetlands, horizontal-subsurface-flow CWs, etc. (Table 6), and the removal rates of horizontal-subsurface-flow CWs were higher than those of constructed floating wetlands. Nevertheless, the application cost of constructed floating wetlands is less compared with that of horizontal-subsurface-flow CWs. Second, the wastewater generated from agricultural production, such as agricultural runoff, dairy wastewater, swine wastewater [99], and other farm wastewater, must be pretreated before being discharged into the environment. Organic loads, N, P, total solids, fat, oil, grease, and pathogens are all common primary-constituent contaminants in agricultural wastewater [100]. Table 6 shows the combination of agricultural wastewater treatments by different plant combinations, adsorption devices, and CWs. Moreover, strategies and techniques for the removal and reuse of phosphorus from agricultural surface runoff using CWs, and the use of CWs as an effective practice for agricultural-runoff management and landscape enhancement, are proposed [101,102]. Third, CWs play a significant role in the removal of contaminants from industrial wastewater, such as those from rice mills, glass factories, and tanneries [103]. The main characteristics of industrial wastewater are its high organic load and COD, high acidity or alkalinity, colour, turbidity, nutrient load, TSS, salinity, colloids, and specific toxic pollutants [104]. CWs efficiently remove the COD and BOD from the water, which results in a reduction in the organic matter and colour turbidity (Table 6).

**Table 6.** Research related to the impact pathways of CWs on water management.

| Rank | Water Body | Plants | Type of CW | Removal Rate |
|---|---|---|---|---|
| | | Stormwater Management | | |
| 1 | Stormwater runoff | *Iris ensata var. spontanea* | Horizontal-subsurface-flow CWs (HSSFCWs) | TSS (75.1%), organics (57.2%), nutrients (50.5%), heavy metals (46.8%) [105]. |
| 2 | Stormwater runoff | *Typha latifolia, Hydrilla verticillate, Eichhornia crassipes, Spirogyra* | Constructed-wetland system integrated with aquatic macrophytes | Faecal coliform (68%), particulate phosphorus (72%), TP (42%), TN (35%), Zn (23%) [106]; TSS (84.3%), COD (79.2%), TN (53.5%). |
| 3 | Stormwater runoff | *Phragmites australis* | Horizontal-subsurface-flow CWs (HSSFCWs) | $NH_4^+$-N (56.5%), $NO_3^-$-N (76.5%), TP (29.5%), Zn (67.2%), Cu (73.2%), Cr (41.7%), Cd (7.1%), Ni (44.1%), Pb (60.6%) [107]. |
| 4 | Agricultural runoff | *Pontederia cordata* | Floating treatment wetland (FTW) | TP (90.3–92.4%), TN (84.3–88.9%) [108]. |
| 5 | Agricultural runoff | Cattails | constructed wetland treatment System with granulated activated carbon | Integrated system's average pesticide concentration (52%), nitrate (61%), phosphate (73%), turbidity (90%) [109]. |
| 6 | Urban stormwater | *Phragmites australis* | Vertical-subsurface-flow constructed wetland (VFCW) | COD (86.54%), TN (89.46%), $NO_3^-$-N (95.87%), $NH_3$-N (80.88%) [110]. |
| 7 | Urban stormwater | *Phragmites australis* | Constructed floating wetlands (CFWs) | PFOA (53%), PFOS (42%) [111]. |
| | | Ecological water | | |
| 1 | Wastewater-treatment-plant effluent | *Phragmites australis* | Combined tidal- and subsurface-flow constructed wetland (TF-SSF-CW) | DOC (88%), DON (91%) [112]. |
| 2 | Slightly polluted river water | Iris, thalia, reed, lotus, Myriophyllum | Three-stage surface-flow constructed wetlands | $NH_4^+$-N (38.4%), $NO_3^-$-N (22.3%), TN (29.1%) [113]. |
| 3 | Urban river | *Cyperus alternifolius* | Horizontal-subsurface-flow CWs (HSSFCWs) | COD (56.18%), TP (61.97%) [114] |
| 4 | Reservoir-type water source | Goosegrass, sedges, water grasses, Polygonum hydropiper bagen, reeds, bulrushes | Ecological floating bed | $BOD_5$ (84.76%), COD (57.14%), Max TN (86.76%), $NH_3$-N (83.78%), $NO_3^-$-N (89.26%), TP (94.02%), TDP (95.89%) [115]. |
| | | Wastewater treatment | | |
| 1 | Domestic sewage | *Typha domingensis Pers* | Constructed floating wetland | COD (55%), $BOD_5$ (56%), TSS (78%), total Kjeldahl nitrogen (41%), $NH_3$-N (38%), TP (37%) [116]. |
| 2 | Domestic wastewaters | *Phragmites australis* | Horizontal-subsurface-flow constructed wetland (HF-CW) | COD (97.8%), $BOD_5$ (92.7%), TSS (97.5%), TN (91.5%), TP (96.9%) [117]. |
| 3 | Dairy wastewater | *Eichhornia crassipes* | Floating constructed wetlands | BOD (86.4%), TS (64.3%) [118]. |
| 4 | Wastewater-treatment-plant tail-water | *Phragmites australis, Typha orientalis Presl, Lythrum salicaria* L., *Acorus calamus* L., *Sagittaria trifolia* L., *Iris wilsonii* | Integrated vertical-flow constructed wetland | COD (40.05%), $NH_4^+$-N (45.47%), TP (62.55%), TN (55.53%), TSS (57.20%) [119]. |
| 5 | Glass-industry wastewater | Pampas grass | Horizontal-subsurface-flow constructed wetland | $BOD_5$ (90%), COD (90%), TSS (99%), TN (95%), TP (96%) [120]. |
| 6 | Tannery wastewater | Common reeds | Horizontal subsurface flow | COD (82%), NH4$^+$-N (96%), Cr (99%) [121]. |

## 6. Discussion

The bibliographic search in this paper reports on the use of CWs as a water-management practice in rural, agricultural, industrial, and urban areas. This paper seeks to identify and analyse problems and to summarise patterns in the use of CWs in each region. Furthermore, the different ways in which CWs are used in different regions are condensed. Thus, the focus is on finding the involvement of CWs as a water-management approach in the water-management process in urban areas. Therefore, the focus of this paper is on the use of CWs in urban water management.

### 6.1. Mechanisms of the Impact of CWs on Water Management

The utilisation of decentralised, integrated, or multifunctional physical water infrastructures is a key aspect of sustainable urban water management. Nancey Green Leigh 's study showed that, in the urban water cycle, the replacement of an old centralised water supply allowed for the diversification of water-supply options and, in so doing, expanded the water cycle within the urban system [122]. CWs can be involved in sustainable urban water management as a green infrastructure [123]. Multiple institutional and technical barriers hinder the transition to decentralised water technology, but we could try to test decentralised water systems on a small scale in practice. Thus, as illustrated in Figure 11b, CW technology could be utilised as a decentralised green infrastructure to participate in stormwater management, wastewater treatment, and ecological water purification in water management. Simultaneously, in the decentralised water-supply-system cycle, CWs are hubs that form several decentralised water-supply modes: agricultural wastewater—water (containing N, P)—farmland irrigation; rainwater collection—water (containing N, P)—nursery irrigation; water collection—clean water—domestic water, etc. [124]. Moreover, CWs have a role in ecological water purification and, consequently, in enhancing the natural water-cycle quality. CWs affect decentralised water-supply systems directly or indirectly by increasing rainwater storage, widening wastewater-treatment pathways, and enhancing ecological water purification, which further influence water-management sustainability.

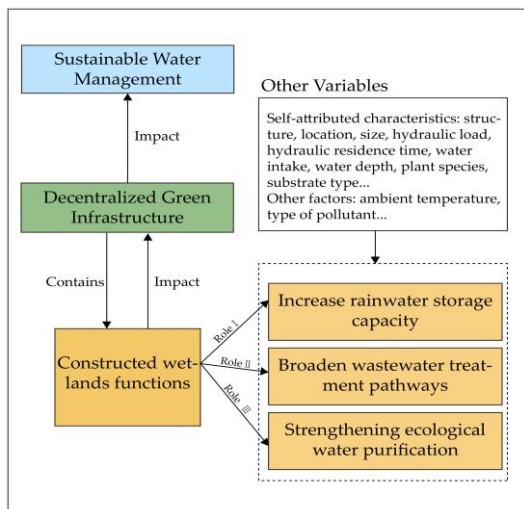

**Figure 11.** Mechanisms for the impact of CWs on urban water management.

### 6.2. Factors That Influence the Role of CWs in Water Management

Sedimentation, filtration, adsorption, plant uptake, and microbial decomposition are all part of the water-management process [81]. The primary classification is based on the sorts of macrophytes utilised in the wetland: free-floating plants, floating leaved plants, emergent plants, and submerged plants. Moreover, CWs can be classified according to the water flow: surface-flow CWs, vertical-flow CWs, and horizontal-flow CWs [97,125], which, in turn, expand on enhanced CWs and hybrid CWs in order to increase the adaptation of

CWs to their location and improve their ability to perform water management. Figure 12 depicts the various types of CWs, as well as their most common appearances.

For water management, various types of CWs are employed in various ways and forms. For example, K Aristeidis et al. used small earthen dams to utilise runoff by intercepting and accumulating all the water in a reservoir, and they show that this is also an important measure for water management [126]. Traditional CWs, such as surface-flow CWs and submerged CWs, are channel-type impermeable structures that are filled with substrate and planted with wetland plants or macrophytes, through which the water flows [127]. Nevertheless, constructed floating wetlands (CFWs), floating treatment wetlands, artificial floating beds/wetlands, ecological floating beds, floating-plant-bed systems, integrated floating systems, integrated ecological floating beds, etc. [128], which are adapted to water-level fluctuations and treated by the direct nutrient absorption by vegetation and increased sedimentation with the aid of biofilm growing on plant roots, are the latest innovations in CWs for water management [129]. On the one hand, both CWs and CFWs are commonly used to purify river water and restore the environment, although classic CWs have limits because of the clogged strata and huge footprint. CFW technology, on the other hand, is less engineered and is easily integrated into existing water environments, such as lakes, ponds, dams, and storage ponds, but it is slightly weaker in terms of the pollutant-removal efficiency [130]. Moreover, CWs are further realised with newer applications, such as bio-electrochemical-assisted constructed wetlands (MET land) [131], green roofs + constructed wetlands [132], and vertical green walls + constructed wetlands [133]. Thus, it is important to flexibly grasp the characteristics of several types of CWs to further explore the deformation of CWs and apply them to urban water management in the future.

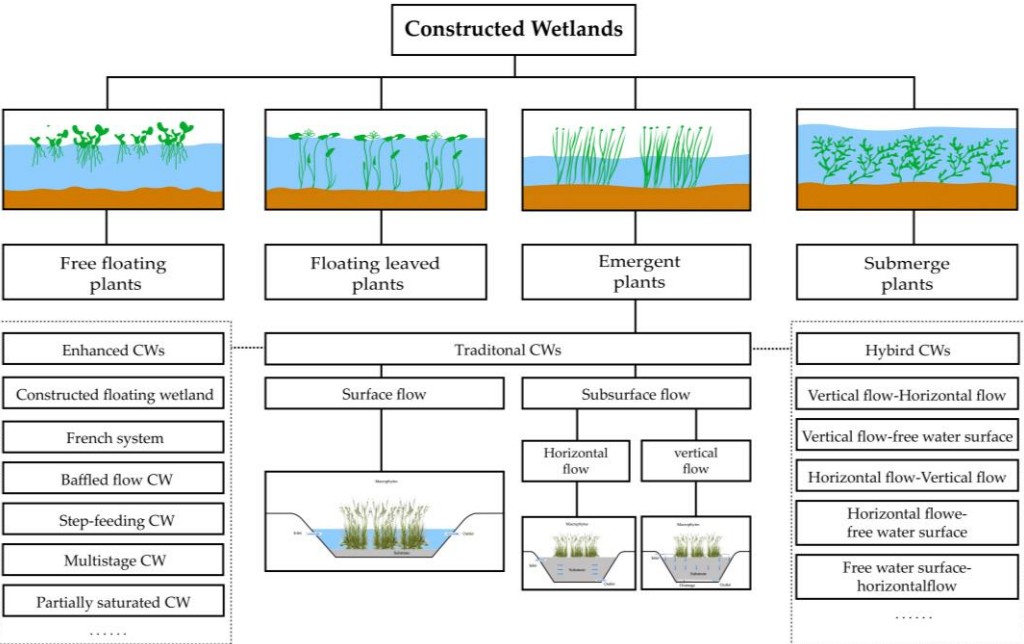

**Figure 12.** Main types and classifications of constructed wetlands [82,83,134–136].

The design and maintenance of CWs for water management, such as nitrogen-removal processes, biodegradation, adsorption, and assimilation, are all dependent on the CW's efficiency [137,138]. First, the design parameters primarily include the substrate, plants, hydraulic retention time (HRT), water depth, aeration, and other parameters; this could be performed by combining different CW structures, or by changing the wetland parameters [139,140]. Current studies show that the main functions of the matrix are the filtration and interception of larger particles and pollutants, such as the total suspended solids, which are primarily removed by matrix filtration and sedimentation [141], and the adsorption of different pollutants [142], which provides electronically accelerated denitrification [143].

Additionally, superabsorbent polymers, recycled concrete aggregates, manganese oxides, and iron oxides will be investigated as CW substrates in the near future [144–146]. In the case of plants, the bacteria attached to the plant's rhizome in the process, as well as near the roots, carry out oxygenation, along with the absorption of pollutants in the water [123]. Among them, aquatic macrophytes are an inaccessible part of CWs [147], and the commonly utilised macrophytes are Oenanthe javanica., Iris pseudacorus Linn., Acorus calamus Linn., Thalia dealbata Link., and Typha angustifolia Linn. Qi Yu et al. demonstrated that the addition of artificial macrophytes significantly improved the COD and N removal in surface-flow constructed wetlands [148,149]. In addition, it increased the influent nitrogen concentration and extended the HRT by increasing the water depth to promote the accumulation of microbial communities and improve the nitrogen-purification capacity [150]. According to Saeed T et al., horizontal-flow wetlands with shallow water depths facilitate simultaneous aerobic and anaerobic removal and are very successful wastewater-treatment systems [151]. Cristina Ávila et al. discovered that continuous and intermittent aeration had a similar efficiency for drug removal, and considering the energy cost, intermittent aeration would be the best choice [152].

The focus of CW management is on substrate clogging and plant harvesting. The ease with which CWs clog has an important influence on the water management, and Wang H et al. demonstrated that the use of substrates, such as an appropriate particle size and multilayer substrates, helped to slow CW clogging [153]. Moreover, selecting aquatic plants with well-developed root systems and the timely removal of leaves and dead roots can significantly reduce CW clogging. Thus, the use of cycles of different substrates needs to be further evaluated in the future and regularly managed. Meanwhile, the amount and frequency of plant harvesting has a crucial influence on the long-term role of CWs, and harvesting and regenerating wetland plants on a regular basis is necessary to completely remove the pollutants absorbed in the system, thereby avoiding further pollution problems due to the reintroduction of nutrients into the water.

*6.3. Additional Benefits of CWs for Water Management*

CWs have the ability to modify the urban environment and provide urban ecological services in addition to stormwater management, wastewater treatment, and urban ecological water purification [154]. For example, the purification efficiencies of CWs designed with ornamental plants is 93.8% for TN, 80.0% for dissolved organic carbon (DOC), 84.0% for the biochemical oxygen demand (BOD$_5$), 77.0% for the chemical oxygen demand (COD), and 99.7% for the turbidity, which not only effectively remove nutrients and improve the wastewater quality, but the cannas also bring certain landscape benefits [155]. In addition, Sergio Zamora et al. used three ornamental plants (*Canna indica*, *Cyperus papyrus*, and *Hedychium coronarium*) as a phytoremediation process for wastewater, and they found that all three plants were able to remove more pollutants than the experimental setup without plants, which confirms that ornamental vegetation can be used in CW systems [156]. Immediately after, LC Sandoval-Herazo et al. investigated the ability of ornamental plants to remove pollutants at different densities, and they showed that high-density planting facilitated the removal of 10% to 20% of the pollutants [157]. José Luis Marín-Muñiz et al. further explored the effects of ornamental-plant monocultures and mixotrophs on domestic constructed wetlands (DCWs), and they found that the use of multiple mixed plants (*Canna hybrid*, *Alpinia purpurata*, and *Hedychium coronarium*) in DCWs provides multiple benefits for water purification, such as: aesthetics, enhanced biodiversity, and the removal of pollutants [158]. Moreover, urban wetlands can mitigate urban heat, store groundwater, perform soil remediation, and provide the social benefits of green spaces in urban environments [159]. CWs can even serve as carbon sinks, accumulating and storing carbon-containing molecules in the form of biomass. The plants and microorganisms in a CW form a consortium that increases the plant biomass while performing water management. Therefore, plants use nutrients for growth and photosynthesis, during which $CO_2$ is eventually converted into biomass, which thereby decreases carbon dioxide emissions [160].

### 7. Conclusions

Using CiteSpace bibliometric visualization software to visualise and analyse the research fields of water management and CWs from 2002 to 2022, this study focused on integrating the literature linked to water-management and CW research as the data source. First, a preliminary analysis of the development trend between water management and CWs was conducted based on the literature postings in the field of water-management and CW research, along with the cooperative-network mapping of countries and authors. Secondly, based on the analysis of the keyword-co-occurrence mapping, keyword-clustering mapping, keyword-timeline mapping, keyword-mutation mapping, literature-citation-co-occurrence mapping, and literature-co-occurrence-timeline mapping in the research field, this paper discusses the research hotspots and frontiers over the last 20 years. In this way, the overall development of the field is evaluated to grasp the changes in the patterns, to discover the future development trends of the research field, and to provide references for this research field.

This research topic involves 125 main research countries and regions from a country/region perspective. The United States is the leading research country in this field by virtue of its literature output, followed by China and Australia. Of the 125 countries/regions involved, the most influential country/region is Switzerland, followed by the United States and Germany. MIKLAS (10), ROBERT (6), and RORY (5) have the most publications in the author-collaboration-network mapping, but their influence is still lacking. There are 23 clusters in the literature-co-citation mapping, and other related studies strongly suggest co-citation relationships. In terms of the research topics, the keyword-clustering mapping of water management and CWs indicates a total of 18 clusters in this field, reflecting the hot topics over the past 20 years. The research related to the research field is classified according to the research object, content, method, etc., and it is divided into four categories: the research methods used in the water-management research of wetlands and CWs, as well as the research content targeted; the function and role of CWs; the water-quality indicators utilised to quantify the role of CWs for water management; the types of CW treatments of water bodies. Each cluster has obvious research characteristics that fully reflect the research hotspots in water management and CWs.

This paper examines water-management and CW research, and compares the research histories, statuses, and trends. We found that CWs focus on water management in three ways: rainwater management, wastewater treatment, and ecological water purification. Moreover, CWs have the potential to become part of decentralised urban-water-supply systems as green infrastructure and to thus participate in urban water management. Furthermore, the influencing factors of CWs for water management, as well as their additional benefits, are discussed, which demonstrates that CWs must be designed and maintained in future research, and must be more involved in water management so that it can be made sustainable through CWs.

The research areas of water management and CWs are compared in this publication. In fact, this is only a small portion of the extant study, and it is far from complete. In the process of realizing CWs for water management, it is worthwhile for researchers to further think about the macroscopic arrangements and microscopic updates of CWs. In future research, the relevant content might be more extensive and in depth, and it needs to be refined for different water qualities and regions. With attention to these details, researchers should place their focus on the following points:

- Enhancing the renewal and use of CWs. CWs serve as water collectors and purifiers, and, as decentralised green infrastructures based on natural solutions, they can participate in the treatment and distribution of decentralised urban-water-supply systems, making urban water management sustainable through rainwater management, wastewater treatment, and ecological water purification, and bringing positive impacts to urban ecosystem services. For starters, typical CWs demand more land-use space; hence, there have been few attempts to incorporate CWs into urban water management in densely populated areas. In the future, CWs, as green decentralised water-supply

systems in urban-water-management practice, will need to "see the needle" typed into urban water management and CW innovation, so that they are integrated into the city, such as highways, rain gardens, and residential areas, and to seek more suitable high-density cities in form and structure that include roof wetlands or green-wall wetlands, which have been used in closed communities for water purification and recycling. Second, in terms of the CW clogging and mechanism aging, update the design parameters and try more substrates, plants, shape combinations, and so on, in order to explore the best paradigm of CWs for urban water management. Moreover, for the natural formation of "accidental" wetlands in the city, use the appropriate CW-related design parameters to design and use them;

- Focus on the monitoring and evaluation of CWs. First, integrating developing technologies necessitates the real-time monitoring of CWs, as well as digital water-quality monitoring, in order to prevent the conversion of CWs from purification to discharge, and from carbon sink to carbon source. Secondly, the relationship between biomass and water purification was studied, and the best time for biomass harvesting was sought. Simultaneously, the performance of the CWs was assessed to quantify their impact efficiency in stormwater management, wastewater treatment, and urban ecological water purification, and to further measure the benefits generated by CWs in urban water management;

- Combine the function and landscape benefits of CWs. There are various landscape plants, but only a few are utilised in CWs, and prior research has focused on the benefits of plants for water purification while overlooking their aesthetic features. Future research must try balancing the water-management benefits of CW plants with the landscape benefits to not only increase the public acceptance of CWs as a significant method of water management, but also to increase the public participation in the maintenance of CWs, which leads to longer lifespans, as well as more beautiful urban landscapes.

In summary, the multiple impacts of CWs on water management have been confirmed in various aspects. However, how to properly employ CWs to promote sustainable water management is a topic that researchers and the general public should investigate further in the future for future research references.

**Author Contributions:** Y.Z. and J.D. provided the research idea and purpose of this study; Y.Z. and J.D. designed the research; Y.Z. and S.H. collected and analysed the data; Y.Z. wrote the paper; Y.Z. supervised, corrected, and revised the paper; X.Y. and M.W. corrected the article language and made some suggestions. All authors have read and agreed to the published version of the manuscript.

**Funding:** This research received no external funding. This study was funded by the Forest Park Engineering Technology Research Center of the State Forestry Administration (PTJH15002), and by the Wuyishan National Park Research Institute Special Project (KJg20009A).

**Institutional Review Board Statement:** Not applicable.

**Informed Consent Statement:** Not applicable.

**Data Availability Statement:** The data used to support the findings of this study are available from the corresponding author upon request.

**Conflicts of Interest:** The authors declare no conflict of interest.

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
