# Peer review of "Knowledge Atlas on the Relationship between Water Management and Constructed Wetlands—A Bibliometric Analysis Based on CiteSpace"

_sustainability, doi:10.3390/su14148288_

Round 1
Reviewer 1 Report
The main objective of the work and the discussion is about the issue of urban wastewater management, but the bibliographic search was not limited to urban regions, or they do not clarify it in the methodology. The authors should clarify if the bibliometric analysis only refers to the urban region or if the bibliographic search also reported the use in rural, agricultural or industrial regions. The authors must synthesize the information in such a way that they can attract the interest of their readers. Specific observations are presented in the attached file.
Reviewer 2 Report
Dear Editor.
I have finished my review on the proposed paper “Knowledge Atlas on the Relationship between Water Management and Constructed Wetlands—A Bibliometric Analysis Based on CiteSpace” sustainability-1773159-peer-review-v1.
Summary of the manuscript:
In the proposed paper, the authors’ goal is to generate country/region maps, author collaboration maps, and analyze research hotspots and research dynamics by using keywords and literature co-citations based on Web of science (WoS) database. The scope is to summarize and analyze the status of research between water management and constructed wetlands.
Review:
1. Generally, the manuscript presents a very interesting topic and the specific research seems to include some significant points for the research community of this field.
2. The proposed paper is very well written with very good use of English language and scientific style. Except some minor grammatical mistakes and word errors. The authors should check again the paper to correct these minor mistakes.
3. The proposed paper is very well structured. It begins with an analytical Introduction with the appropriate references that helps the reader to get into the subject immediately. In Introduction there is an effort to provide previous studies with similar scientific content, which took place in the research area and in other countries. I proposed to add the following research which is relevant to your study and you can find it in SCOPUS [Kastridis and Stathis 2015. The effect of small earth dams and reservoirs on water management in north Greece (Kerkini municipality). Silva Balcanica, 16(2), pp. 71-84, https://silvabalcanica.files.wordpress.com/2015/09/sb_162-2015-071-084.pdf]. Authors describe and set very well the scientific problem and how other researchers have approached. At the end of Introduction, authors clearly state the goals of the research.
4. The methodology is generally very interesting, and well explained, so other researchers could easily repeat it.
5. The results scientifically explained and are OK.
6. The quality of the work in Discussion is very high.
7. Conclusions are appropriate for this paper.
So the paper is an excellent scientific contribution.
Round 2
Reviewer 1 Report
In this second version, the authors present a notable improvement in the manuscript, after having satisfactorily addressed the suggestions of the reviewers.
I consider it appropriate that the authors mention that in Latin America, and particularly in Mexico, there are some studies at the mesocoms level for household or community water management. I recommend that you add the following citations as an example:
Sandoval-Herazo LC, Alvarado-Lassman A, López-Méndez MC, Martínez-Sibaja A, Aguilar-Lasserre AA, Zamora-Castro S, Marín-Muñiz JL. Effects of Ornamental Plant Density and Mineral/Plastic Media on the Removal of Domestic Wastewater Pollutants by Home Wetlands Technology. Molecules. 2020 Nov 12;25(22):5273. doi: 10.3390/molecules25225273. PMID: 33198195; PMCID: PMC7696903.
Marín-Muñiz, J.L., M.E. Hernández, M.P. Gallegos-Pérez, S.I. Amaya-Tejeda. 2020. Plant growth and pollutant removal from wastewater in domiciliary constructed wetland microcosms with monoculture and polyculture of tropical ornamental plants. Ecological Engineering 147:105658.
Zamora, S.; Marín-Muñíz, J.L.; Nakase-Rodríguez, C.; Fernández-Lambert, G.; Sandoval, L. Wastewater Treatment by Constructed Wetland Eco-Technology: Influence of Mineral and Plastic Materials as Filter Media and Tropical Ornamental Plants. Water 2019, 11, 2344. https://doi.org/10.3390/w11112344.
